# Rapid evolution of pre-zygotic reproductive barriers in allopatric populations

Anjali Mahilkar,[1] Prachitha Nagendra,[1] Pavithra Venkataraman,[1] Saniya Deshmukh,[1] Supreet Saini[1]

**ABSTRACT** Adaptive divergence leading to speciation is the major evolutionary process generating diversity in life forms. The most commonly observed form of speciation is allopatric speciation which requires that gene flow be prevented between populations by physical or temporal barriers, as they adapt to their respective environments. Eventually, these adaptive responses lead to accumulation of mutations in different lines. The increased genetic distance between the lines is known to lead to the populations becoming reproductively isolated. A widely accepted theory is that speciation simply occurs as a by-product of adaptive response of the populations. Several examples of allopatric speciation from ecology and laboratory exist. However, we know little about the nature (pre- or post-zygotic) of barriers that arise first in this process. Understanding the first barriers that arise between populations is key to understanding how the process of speciation initiates. In recent years, fungi have been used as model organisms to answer questions related to the evolution of reproductive isolation. Here, we show rapid evolution of pre-zygotic barriers between allopatric yeast populations. We further demonstrate that these pre-zygotic barriers arise due to altered mating kinetics of the evolved population. Moreover, our non-adaptive evolution experiments with yeast under limited selection pressure also show rapid emergence of reproductive isolation. Overall, our results show that evolution of pre-zygotic reproductive barriers can occur as a result of natural selection or drift.

**IMPORTANCE** A population diversifies into two or more species—such a process is known as speciation. In sexually reproducing microorganisms, which barriers arise first —pre-mating or post-mating? In this work, we quantify the relative strengths of these barriers and demonstrate that pre-mating barriers arise first in allopatrically evolving populations of yeast, *Saccharomyces cerevisiae*. These defects arise because of the altered kinetics of mating of the participating groups. Thus, our work provides an understanding of how adaptive changes can lead to diversification among microbial populations.

**KEYWORDS** evolution, reproductive barriers, adaptation, yeast, pre-mating, post-mating

How diversity of life forms arises on Earth is an open question. Although, Charles Darwin's *Origin of Species* explained natural selection as a mechanism for populations to adapt to prevailing environmental conditions, we do not yet understand the fundamental evolutionary forces leading to reproductive isolation. According to one idea, reproductive isolation, leading to speciation, is thought to evolve incidentally as a by-product of adaptation of populations to diverging selection pressures in allopatry (1, 2). Most of the evidence supporting this hypothesis of speciation being a fallout of adaptive response of the population to divergent selection comes from the study of recently diverged species, although speciation between populations adapting to

Address correspondence to Supreet Saini, saini@che.iitb.ac.in.

The authors declare no conflict of interest.

See the funding table on p. 12.

10.1128/spectrum.01950-23 **1**

identical environment has also been reported and is known as mutation-order speciation (3).

The barriers to gene flow leading to speciation could be pre-zygotic or post-zygotic (4). However, little is known about the relative importance of the mechanisms and forms of reproductive barriers by which speciation occurs. The central challenge that remains in understanding speciation is understanding the nature of and order in which these barriers arise between two populations, eventually leading to speciation (5). Efforts in this direction come from the analysis of closely related species or via laboratory experiments. Populations in allopatry, in the absence of reinforcement or assortative mating, are thought to more readily establish post-zygotic barriers. On the other hand, it is thought that pre-zygotic barriers are more relevant for populations in sympatry (6–8). Overall, conclusions regarding the relative roles of pre- and post-zygotic barriers in initiating speciation process are contentious (5, 6, 8–19).

Experimental evolution, to understand how barriers to gene flow arise, demonstrates that adaptation to ecological niches leads to evolution of reproductive barriers in a relatively short time (20, 21) and can result in pre-zygotic [pre- (20) or post-mating (22)] or post-zygotic (23) barriers. While these experiments have largely been performed with populations with standing genetic variation (21, 22, 24–27), more recently, populations of isogenic yeast have been used as a model system to understand evolution of reproductive isolation (7, 28, 29). This approach allows one to identify *de novo* variants that are potentially involved in reproductive isolation (30). However, the relative contribution of pre- and post-mating barriers in the early steps of speciation remains uncharacterized. In some studies, the experimental design precludes asking this question.

In this work, we ask the following question: which barriers arise first between populations in allopatry, evolving under selection and drift? To answer this question, we perform adaptive and non-adaptive evolutionary experiments with the yeast *S. cerevisiae*. Haploid isogenic yeast populations of both mating types (MATs) were evolved in glucose- and galactose-limited growth media for 600 generations, respectively. Three MATa and MATα lines were evolved in glucose, and three MATa and MATα lines were evolved in galactose—thus, making a total of 12 allopatric populations in the experiment (Fig. S1). To study evolution in the absence of selection, 44 independent lines of haploids (22 lines of MATa and 22 lines of MATα) were also evolved in a mutation accumulation experiment for 70 transfers (~1500 generations). We demonstrate that metabolic specialization in the adaptive evolution experiment and the action of drift alone in the non-adaptive experiment leads to rapid evolution of pre-zygotic barriers, in the form of reduced mating efficiency as a result of altered mating kinetics, where the mating efficiency of different strains as a function of time varies quantitatively from each other. We demonstrate that, in specific environmental contexts, acquisition of a few SNPs only can establish significant barriers to mating. Overall, we demonstrate that evolution of reproductive barriers can start with pre-zygotic barriers among populations in allopatry.

## RESULTS

### Adaptive response in glucose and galactose

During allopatric evolution, an increase in fitness was observed due to a decrease in the lag phase duration and an increase in the growth rate (Fig. 1A and B; Table S1). The adaptive path of the galactose-evolved lines follows a distinct phenotypic path. In the first 200 generations, the evolved lines exhibit a reduction in the lag phase duration and an increase in the growth rate. In the next 200 generations (200 to 400 generations), the evolved lines exhibit a statistically insignificant increase in both, the growth rate and the lag phase duration. In the last 200 generations of our experiment (400 to 600 generations), a decrease in the lag phase duration was observed. However, this decrease in the lag phase duration was observed at a fitness cost of a decrease in the growth rate.

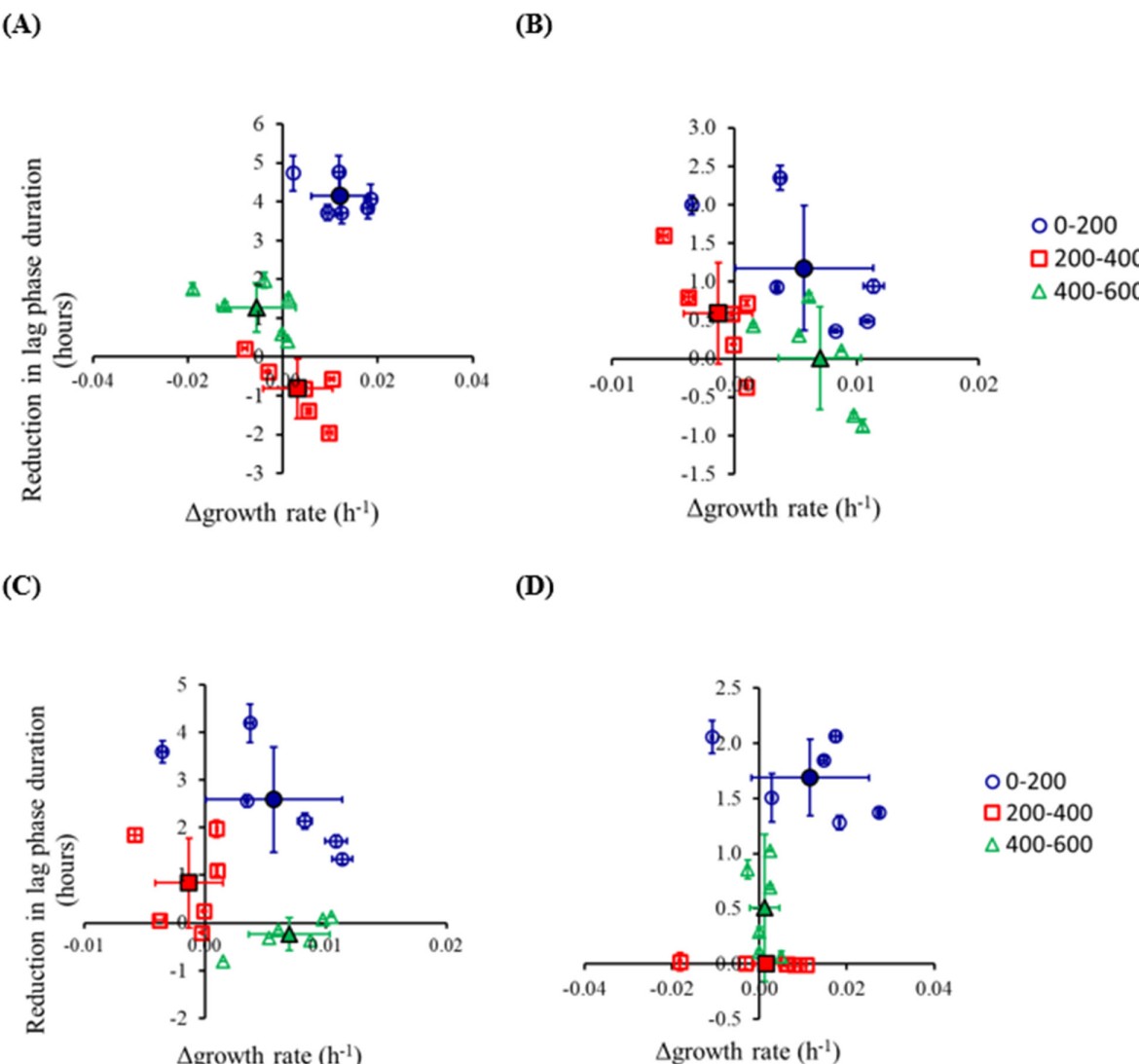

**FIG 1** Adaptive response of haploid yeast during evolution in low galactose (A) and glucose (B). Changes in growth rate and lag phase duration every 200 generations are shown. Adaptive response of haploid yeast, in terms of changes in growth rate and lag phase duration, for (C) galactose-evolved lines in glucose and (D) glucose-evolved lines in galactose. Open symbols are independent lines. Closed symbols have an average of the six lines. Growth experiments were performed three independent times. The average and standard deviation are shown. The adaptive response (growth rate and lag phase duration) of the populations for every 200 generation interval was compared. Other than the comparison of lag phase reduction for the glucose-evolved lines, all differences were statistically significant; $P < 0.005$, two-tailed $t$-test Welch. See Table S1 for all $P$-values.

As compared with the galactose-evolved lines, the glucose-evolved haploid lines exhibit a distinctly different pattern. Adaptation in glucose was characterized by an increase in growth rate and decrease in lag phase duration in the first 200 generations. In the next 200 generations, a decrease in lag phase duration was observed. In the final 200 generations of the adaptation experiment, an increase in growth rate was observed.

The growth characteristics of the evolved lines when studied in the other environment (i.e., galactose-evolved lines in glucose) are as shown in Fig. 1C and D; Table S1. Adaptation in glucose and galactose increases fitness in galactose and glucose, respectively. Adaptation in other carbon sources, when a population has been evolved in a particular carbon source, has been previously reported and is thought to occur due to adaptation of glycolysis as a process (31, 32).

## Pre-zygotic barriers to mating evolve rapidly

Reproductive barriers between populations can be quantified via studying three processes in yeast: change in mating efficiency, mitotic growth of the hybrid, and meiotic efficiency of the hybrid. The pairwise mating efficiencies between all evolved haploid lines exhibit a marked decrease, as compared with those of the ancestor (Fig. 2; Fig. S2). This trend is observed for both, ecological speciation and mutation-order speciation. The decrease in the mating efficiency is most rapid between the glucose-evolved lines.

The decrease in mating efficiency between evolved haploids could possibly be due to one of two reasons. First, the mitotically evolving lines diverged genetically and as a result, despite retaining intact mating pathways, no longer mate with each other with the same efficiency. Second, since these lines were evolved without selection pressure to retain the genes necessary for mating, these lines acquired mutations, which rendered them incapable of mating. To distinguish which one of these two possibilities play out in our evolved populations, we switched the mating type of each of the haploid lines at 600 generations. For example, the mating type of galactose-evolved **a** was switched to **α**. A similar mating type switch was done for all haploid lines. We then quantified the mating efficiency of each evolved haploid with itself (carrying the opposite mating type). Each of the evolved haploids mates with their mating type switched counterpart with considerably higher efficiency, as compared with the average mating efficiency of the evolved haploid with all other evolved haploid lines (Fig. 3). These results clearly indicate that the evolved haploids retain intact mating pathways. Lines glu1a and glu2a were left out from this analysis, as these lines had undergone an autodiploidization during the course of the experiment (Fig. S3) (33, 34).

From the 12 evolved haploid lines and the ancestor **a** and **α**, 49 hybrids could be possibly generated. Of all these possibilities, we were only able to create 42 (Fig. S4). Seven hybrids, all from the glu1a and glu2a lines, could not be created. Thereafter, we characterized the growth kinetics of the 42 hybrid lines in glucose and galactose environments.

## Nature of epistasis between beneficial mutations in glucose and galactose adapted lines

The hybrids were separated into five groups. Those resulting from mating between (i) two haploids evolved in glucose (glu-glu), (ii) haploids evolved in galactose (gal-gal), (iii) a glucose- and a galactose-evolved haploid (glu-gal), (iv) a glucose-evolved haploid with an ancestral haploid (glu-anc), and (v) a galactose-evolved haploid with an ancestral haploid (gal-anc). The performance of all five groups of hybrids was compared with that of the ancestral diploid.

In glucose, the hybrids in the glu-glu, glu-anc, and the glu-gal group all exhibited a decrease in the growth rate, compared with the ancestral diploid (Fig. 4A and B; Fig. S5; Tables S2 and S3). This growth defect is most severe in the glu-glu diploids. Thus, bringing together adaptive mutations in a single genome has a detrimental effect on cellular fitness. This effect has previously been seen during growth in glucose; however, the precise mechanistic details remain unknown (35). The fitness effect of epistatic interactions on yeast is known (36, 37), and its possible role as a mechanism leading to speciation has been suggested recently (38). On the other hand, the hybrids exhibit a qualitatively different pattern, when grown in galactose. The gal-gal group of hybrids exhibits the greatest growth rate and among the shortest lag phase duration. These results suggest that epistasis between beneficial mutations in glucose is qualitatively different from that in galactose. In addition, a small number of hybrids exhibit a statistically significant decrease in the meiotic efficiency, as compared with the ancestor (Fig. S6).

Overall, our data suggest that, in allopatry, pre-zygotic barriers arise significantly faster as compared with post-zygotic barriers. In the framework of our evolution experiment, adaptation has two components: (i) decrease in lag phase duration and (ii)

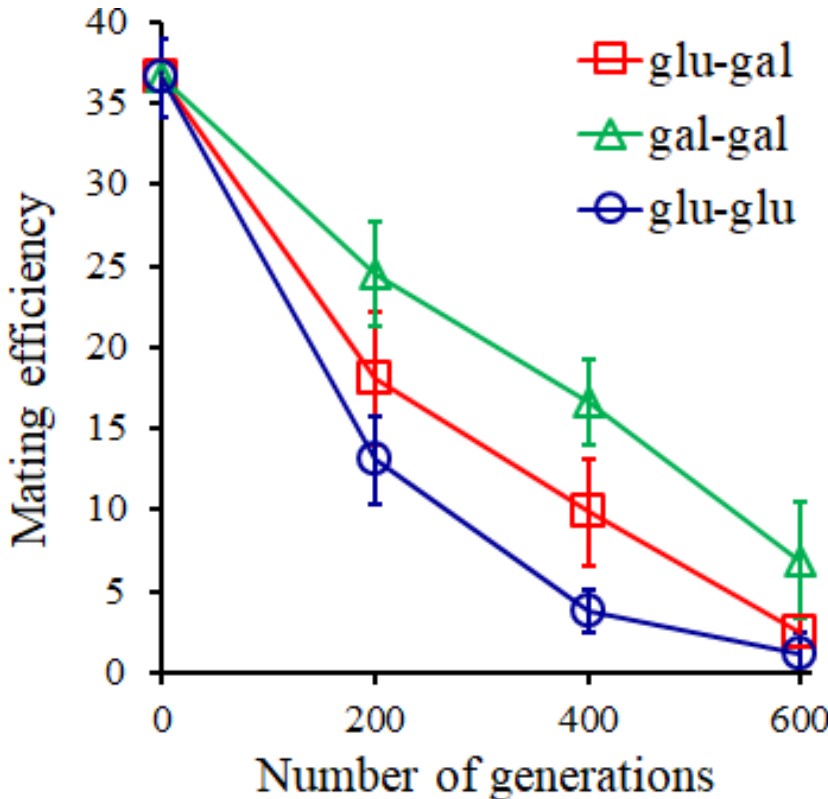

**FIG 2** Mating efficiency between the evolved lines evolved in glucose or galactose environments. The data represent the average of nine possible pairings for each condition and time point. All experiments were performed three independent times. The average values and the standard deviation are shown in the figure.

increase in growth rate. No correlation exists between these values of components of fitness and that of pre- or post-zygotic barriers to gene flow.

### Genome sequencing reveals signatures of convergent evolution

To identify the genetic basis of adaptation and reproductive isolation between the evolved haploid lines, we sequenced the genomes of the 12 haploid lines, after adaptation for 600 generations (Fig. S7;Table S4 and S6), and compared with the ancestral sequence. The sequencing results show evidence of convergent evolution. Two glucose-evolved lines have mutations in MNS1, an ER membrane protein responsible for α-mannosidase activity (a stop codon and a frameshift mutation). MNS1 mutants exhibit a significantly longer life span in yeast (39) and in other organisms (40). Two galactose-evolved lines have mutations in MNL1 (a stop codon and a non-synonymous mutation), another Mannosidase-like protein, which works via formation of a complex with the protein disulfide isomerase, PDI1 (41, 42). Two lines (one glucose and one galactose evolved) have point mutations (both synonymous) in the gene PRM7. PRM7p is induced several fold in the presence of pheromone and is thought to be involved in cell-cell communication (43, 44).

### Evolved lines exhibit altered kinetics of mating

While these similarities in genic targets exist, considerable difference in the mutational targets are also present between the different evolved lines (Table S3). These results lead us to hypothesize that altered mating kinetics in the evolved lines lead to establishment of first reproductive barriers. To test this possibility, we performed mating experiments where kinetics of mating between evolved haploids **a** and their counterparts **α** (obtained

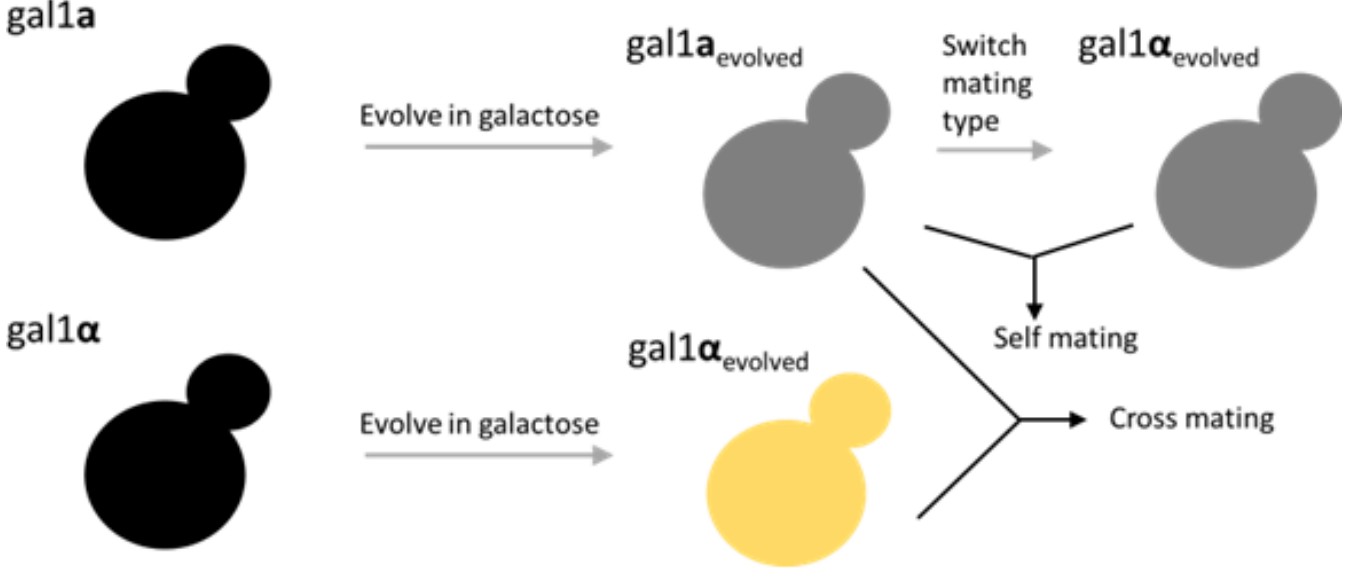

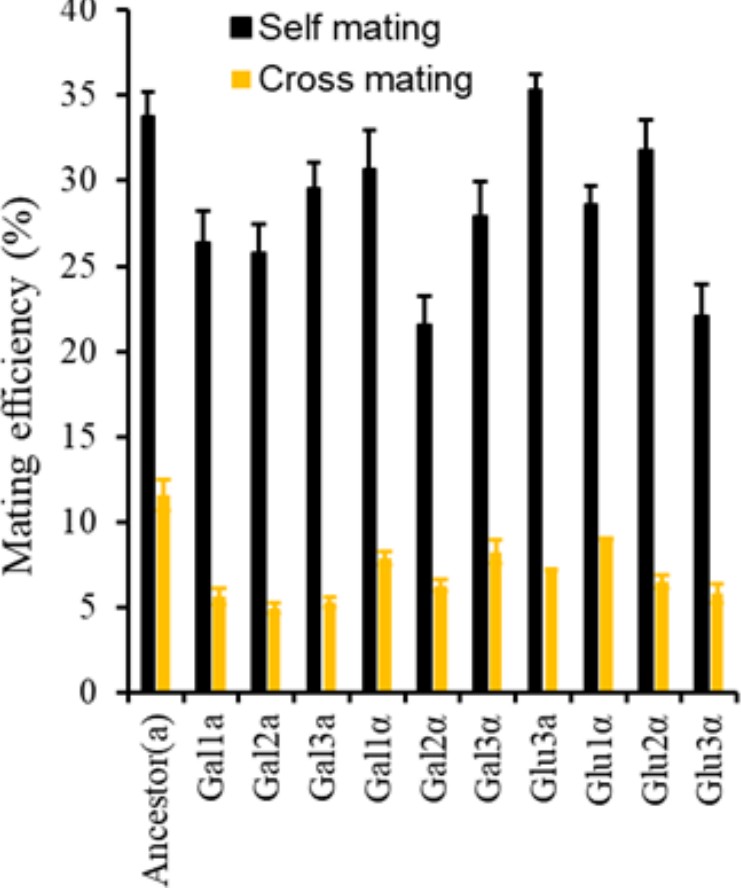

**FIG 3** (Top) Schematic of cross- and self-mating of an evolved haploid. (Bottom) Mating efficiency of the evolved haploids with their mating switched (black bars). Mating efficiency of an evolved haploid with itself (mating type switched) is greater than mating efficiency of other haploids of the opposite mating type (yellow bars). Cross-mating efficiency represents the average of the mating efficiency with all other six evolved haploids of the opposite mating type and the ancestor. All experiments were performed three independent times. The average and the standard deviation are presented in the figure.

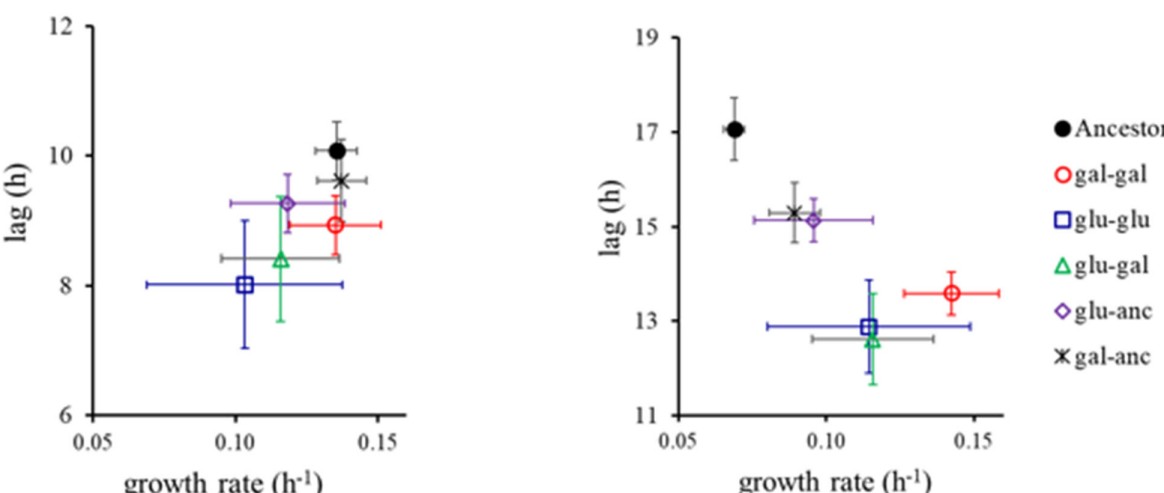

**FIG 4** Mitotic performance of the hybrids in glucose (A) and galactose (B). The hybrids are separated in five groups, depending on the ancestral haploids. Average and standard deviation of three independent experiments are shown. See Tables S1 and S2 for statistics between the five groups and Fig. S5 for individual data points.

by switching their mating type). As shown in Fig. 5 and Fig. S8, the evolved haploids, when mated with their opposite mating type (obtained by switching mating type), exhibit (i) no mating defect and (ii) altered mating kinetics. This observation lends further strength to the argument that reproductive barriers between the haploid-evolved lines (and the ancestor) are largely a result of the altered program of the mating kinetics for a cell.

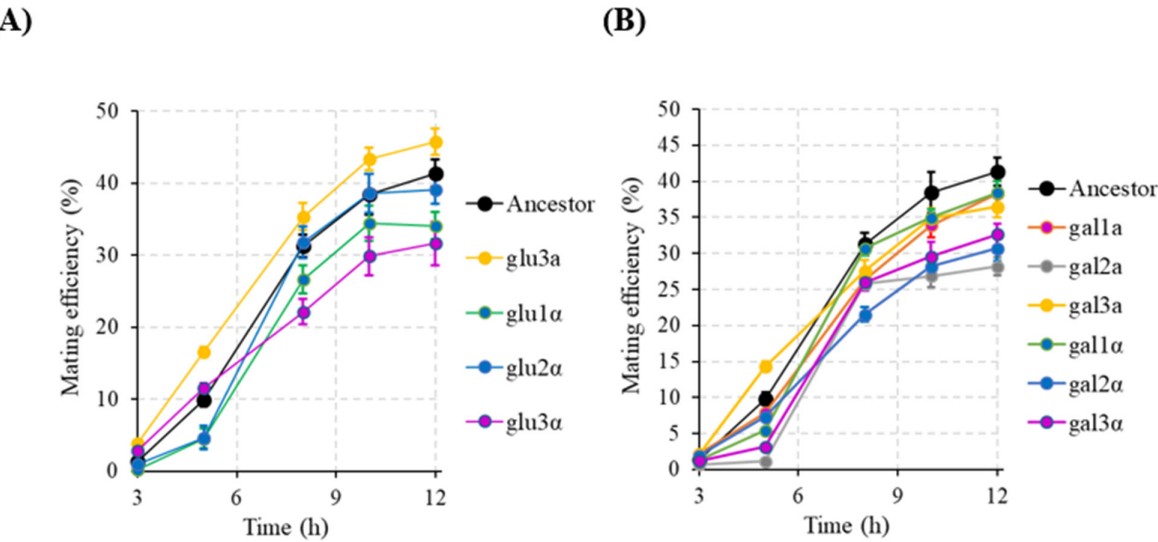

**FIG 5** Kinetics of mating efficiency between (A) glucose-evolved haploids and the switched mating partner and (B) galactose-evolved haploids and the switched mating partner. Ancestor mating efficiency data are between the ancestral **a** and ancestral **α**. All experiments were performed in triplicate. Average values and the standard deviation are reported. In (A), other than glu2α, all profiles are statistically different from those of the ancestor ($P < 0.05$). In (B), all profiles, other than gal1α, are statistically different from those of the ancestor ($P < 0.05$).

## Pre-zygotic barriers arise first in allopatric populations evolved under drift

The null model of speciation suggests that speciation results as a by-product of an adaptive process. Our data above also suggest this. However, this hypothesis has never been explicitly tested in an experimental context. In order to do this, we evolved 22 haploid lines **a** and **α** each in a mutation accumulation (MA) experiment. The 44 lines (a1-a22 and α1-α22) were propagated for 70 transfers (~1,500 generations). Since a severe bottleneck is imposed at every transfer, the role of selection is largely absent and drift dictates evolutionary trajectory. MA experiments with microbial systems demonstrate that, under constant propagation under these conditions leads to a reduction in the fitness of MA lines (45). After 70 transfers, the mean fitness of the evolved lines is ~0.98 times that of the ancestor (Fig. S9).

Our results show that after 70 transfers, a few lines exhibit a statistically significant decrease in the mating efficiency with the ancestor haploid. This decrease in mating efficiency has taken place in the absence of adaptation and is present in varying degrees (ranging from a decrease of ~40% to an increase of ~4%), among the 44 lines evolved. There is no correlation between the adaptive response of the MA lines and the extent of change in the mating efficiency with the ancestral haploid. These results clearly demonstrate that reproductive isolation can, in addition to resulting from adaptation, also result as a by-product of drift (Fig. 6).

While the identity of mutations accumulated in the MA experiment is not yet known, genetic convergence in these lines is unlikely to have happened. MA experiments lead to evolution largely under the action of drift with minimal selection. As a result, the mutational targets' identity is not related to adaptation in the environment of propagation. This leads to the observation that, in an MA experiment, fitness decreases with the number of propagations (46, 47).

## DISCUSSION

Speciation is the fundamental process that generates diversity of life forms on Earth. The most widely accepted explanation for speciation is that it occurs as a fallout of adaptation to diverging selection. Evolution of reproductive barriers among individuals in a population is a classic long-standing interest among biologists trying to understand the process of speciation. However, which barriers arise first as groups begin to diverge in allopatry?

Our data suggest that upon starting with isogenic populations, adaptation in allopatry leads to rapid evolution of pre-zygotic barriers in yeast. In addition, drift alone can also lead to establishment of pre-zygotic barriers. Mating kinetics in yeast are intricately controlled (48). In fact, not only adaptation but the metabolic state of isogenic cells control the effectiveness and selection of a mating partner (49). In context of adaptation to galactose, different ecological contexts lead to non-random segregation of GAL alleles among environmental isolates (50). Additionally, alleles of galactose utilization regulators GAL4 and GAL80 have been isolated in the laboratory (51), leading to altered growth and fitness in different environments (52). In specific environmental contexts, these allelic distributions in an otherwise isogenic background are sufficient to establish a pre-zygotic barrier (Fig. S11).

Reproductive barriers could arise in several ways. Pre-zygotic barriers have been known to arise rapidly, in response to divergent selection (53, 54), although a few negative examples are known too (55, 56). Overall, however, pre-zygotic isolation, as a by-product of adaptation, is observed quite frequently (21). Post-zygotic barriers can be (i) universal (57–59) among closely related species or (ii) environment dependent in nature. Regarding (ii), while negative epistasis is widely observed among beneficial mutations (36, 37, 60), sign-epistasis is relatively infrequent. Antagonism due to a beneficial mutation is also strongly dependent on the choice of the two environments (32, 61, 62). These results suggest that while post-zygotic barriers could arise due to multiple mechanisms and have been predicted to arise in theory too (38), it is likely that pre-zygotic barriers arise rapidly in allopatry due to behavioral changes. In yeast, a

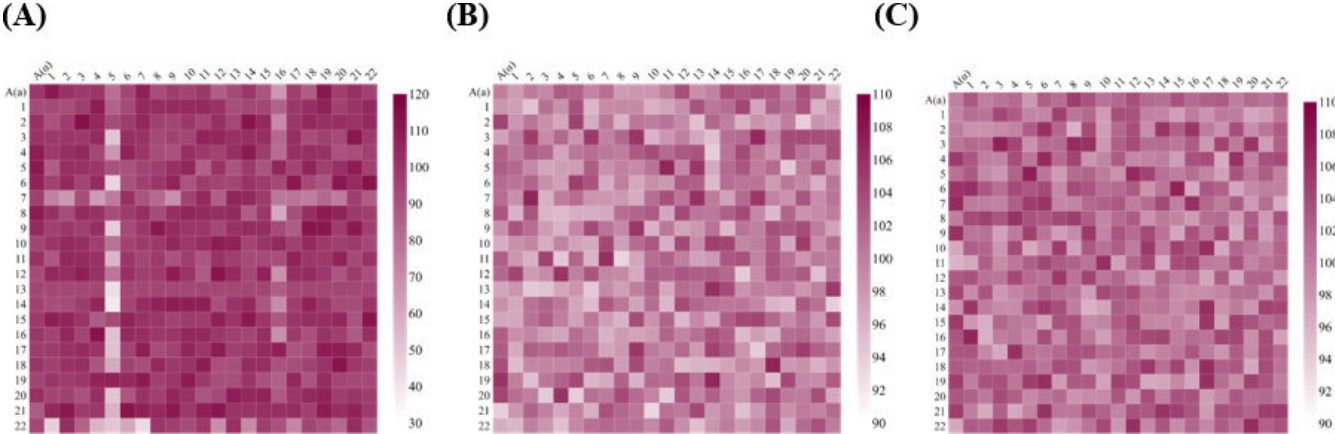

**FIG 6** Reproductive isolation emerges most rapidly via pre-mating barriers for haploids evolving under drift. (A) Mating efficiency of MA lines of α (*x*-axis) and a (*y*-axis). (B) Mitotic growth rate of the hybrids, as compared with the ancestor. (C) Meiotic efficiency of the hybrids, as compared with the ancestor. For (A), (B), and (C), the ancestral mating efficiency, mitotic growth rate, and meiotic efficiency, respectively, are normalized to 100. Numbers "1" to "22" on the *x*- and the *y*-axes represent the 22 evolved haploid lines. "A" refers to the ancestor. All experiments were performed three independent times. The average of the three is reported here. The standard deviation for each data point is less than 10% of the data value. The ancestral mating efficiency (A), mitotic growth rate (B), and meiotic efficiency (C) are normalized to 100 in each of the three plots. Also see Fig. S10 for the three plots with identical range for the heat plot.

laboratory experiment demonstrated evolution of the post-zygotic barrier within a few hundred generations (7). The relative role of pre- and post-zygotic barriers in establishing early reproductive isolation between populations is an open question and also likely dependent on the environmental context in which the allopatric populations adapt (63).

## MATERIALS AND METHODS

### Strain construction

The *Saccharomyces cerevisiae* strain used in this work is a derivative of ScPJB644 (*MATa MEL1 ade1 ile trp1-HIII ura3-52*) (64) with auxotrophic markers inserted near the MAT locus to identify the mating type of the haploids. To prepare the MAT**a** and MAT*α* ancestral population, ScPJB644 diploid was sporulated and dissected to obtain isogenic **a** and *α* haploids from a single ascospore. Two different auxotrophic markers, TRP1 and URA3, were inserted at the same site in ARS314 on chromosome III near the MAT locus on each **a** and *α* type haploids, respectively (65). They are located between genes PHO87 and BUD5, but not disrupting either gene (Fig. S12), as described earlier (66).

URA3 was amplified from the plasmid p426GPD (64) using the primer set pSc011 and pSc012 (all primers listed in Table S5). TRP1 was amplified from the plasmid p424TEF (67) using the primer set pSc014 and pSc015. Both URA3 and TRP1 fragments were further processed to two sequential rounds of PCR to increase the length of flanking ends for efficient recombination, using the primer sets pSc018 and pSc019 and pSc020 and pSc021. Purified PCR products were transformed into **a** and *α* type haploids by electroporation, using an Eppendorf Eporator. The two haploids thus obtained are referred to as ScAM04 (*MATα URA3*) and ScAM05 (*MATa TRP1*). The ancestral diploid was obtained by mating the two, and the resulting strain is referred to as ScAM06 (*MATa TRP1/MAT α URA3*). All strains were grown in YPD [0.5% yeast extract, 1% peptone, and 2% dextrose (wt/vol)] at 30°C and shaking at 250 rpm in 25 × 100 mm test tubes unless specified otherwise. The freezer stocks of all strains are stored at −80°C in 25% (vol/vol) glycerol.

The GAL4c GAL80s-1 strains used in this work are described in (51, 68).

### Evolution experiment

The ancestral populations for the evolution experiment were started from the freezer stocks streaked on YPD plates. Single colonies of each ScAM04 and ScAM05 were

inoculated in 5-mL YPD for 36 hours at 30℃ and shaking at 250 rpm. These cultures were used as inoculum for the evolution experiment. Three replicate lines of each haploid strains were started in 0.2% (wt/vol) glucose or galactose in standard synthetic complete medium (SCM: 0.671% YNB with nitrogen base and 0.05% complete amino acid mixture). Thus, in total, we maintain 12 populations, three of each kind.

Adaptive evolution experiments were performed by serial dilutions every 24 hours in SCM with the appropriate carbon source. Every 24 hours, growing cultures were diluted 1:100 in fresh SCM medium with appropriate carbon source yielding ~6–7 generations every 24 hours. Intermediate generations were cryopreserved in 25% (vol/vol) glycerol every 200 generations.

## Mutation accumulation experiment

ScAM04 was spread on YPD plates for single colonies. Prior to spreading, a small area was identified and marked on the plate. The colony inside (or closest to) the marked area, after 48 hours of growth at 30℃, was suspended in 2-mL PBS buffer. A fraction of this volume was spread on a fresh YPD plate for single colonies, and the process is repeated for 70 transfers. A total of 22 independent lines of ScAM04 were maintained in this experiment. Similarly, 22 independent lines for ScAM05 were propagated for the MA experiment.

## Fitness measurements

Evolved lines and ancestor (a or $\alpha$ haploids) from freezer stocks were revived in 5-mL YPD and incubated for 48 hours and then transferred 1:100 to a glycerol-lactate medium (gly-lac: 3% glycerol, 2% potassium lactate (pH 5.6), 0.671% YNB with nitrogen base, and 0.05% complete amino acid mixture) and incubated for 48 hours at 30℃ and shaking at 250 rpm. Two-mL SCM with appropriate carbon source (0.2% glucose/galactose) were inoculated for a final $OD_{600}$ of 0.01. 150One hundred fifty µL of the cultures was then transferred to a clear flat-bottom 96-well plate (Costar) in triplicates and incubated at 30℃ in an automated microplate reader (Tecan Infinite M200 Pro) until they reached a stationary phase. A gas-permeable Breathe-Easy (Sigma-Aldrich) sealing membrane was used to seal the 96-well plates. $OD_{600}$ measurements were taken every 1 hour with 15 minutes of orbital shaking at 5 mm amplitude before the readings. Growth rates were calculated by plotting log (OD) from the exponential phase of growth against time. The slope of the straight-line fit was determined as the growth rate of the strain. The point of intersection of this straight line, with the straight line with the equation log (initial OD), was taken as the duration of the lag phase.

## Mating efficiency

Mating efficiency between two haploids was calculated as described in (69). Briefly, ancestor and evolved haploid lines were revived in 5-mL YPD cultures from freezer stocks and grown till saturation at 30℃. The haploids of opposite mating type were spot plated on a 1-cm$^2$ area leaving a 1-cm space between the two and incubated at 30℃ for 24 hours. After incubation, cells from either side of the square were scraped and resuspended in sterile water and measured OD600 to determine cell density. Equal numbers of both the haploids were mixed in 1.5-mL sterile microcentrifuge tubes, and 5 µL of the mix was spot plated on a 1-cm$^2$ marked area at the center of the plate and allowed to mate for 7 hours at 30℃. After 7 hours, cells from this area were scraped and plated on YPD plates for single colonies and incubated at 30℃ for 24 hours. The diploids were counted by replica plating on Uracil and Tryptophan double dropout synthetic medium agar plates. A minimum of 500 colonies were transferred to the double dropout plate for quantification of the mating efficiency. Mating efficiency was determined by the following formula:

$$\text{Mating efficiency} = \frac{\text{No. of haploids mated}}{\text{No. of haploids}}$$

## Sporulation efficiency

The hybrids or homozygous diploids, along with the ancestral diploid, were revived by streaking on YPD plates from freezer stocks. Single colonies from each line were patched on to freshly prepared pre-sporulation GNA plates (GNA medium: 5% D-glucose, 3% nutrient broth, 1% yeast extract, and 2% agar) and incubated at 30°C for 24 hours and re-patched again on GNA plates for another 24 hours. Small lumps of cells were then patched onto sporulation medium plates containing 1% potassium acetate and 2% agar and incubated at 25°C for 5 days and 30°C for 3 days. Sporulated/un-sporulated cells were directly counted under ×400 magnification using a microscope to determine the sporulation efficiency (a minimum of 500 cells). The following formula was used to determine the sporulation efficiency.

$$\text{Sporulation efficiency} = \frac{\text{No. of spores}}{\text{Total No. of cells}}$$

## Self-mating efficiency of the evolved lines

Mating type **a**-evolved haploid lines after 600 generations were transformed with a plasmid carrying the *HO* gene to obtain diploids of the evolved lines by selecting on URA dropout plates. The diploids were then sporulated as described previously. Tetrad dissection was performed via standard zymolyase treatment and manual dissection under the microscope. Haploids of both mating types were obtained from a single ascus and identified using colony PCR and primers described in (70). We then performed mating assays as described previously and used colony PCR of the *MAT* locus to determine the number of haploids and diploids to determine the mating efficiency. At least 150 colonies were analyzed using PCR to quantify the mating efficiency between two haploids in a single experiment. Self-mating efficiency reported here is an average of three independent self-mating experiments for all pairs analyzed.

## FACS experiment

A 5-mL YPD culture was started from a single colony on a YPD plate. Saturated cultures were sub-cultured 1:100. When the cultures reached an OD of 1.00, a volume of 1.5 mL of the culture in the exponential phase was spun down ($2–3 × 10^6$ cells/mL) and resuspended in 100-µL autoclaved water. A volume of 1 mL of 70% ethanol was added along the sides of the vial while vortexing. The cells were then incubated at room temperature for an hour and then kept at 4°C overnight. The cells were then washed with 500 µL of RNase buffer (0.2M Tris-Cl with 20-mm EDTA, pH 8.0) and resuspended in 100 µL of the same buffer. RNase A was added to a final concentration of 1 µg/µL. The cells were then incubated at 27°C for 4 hours and then washed with 1-mL PBS (centrifuged at maximum speed for 45 sec) and resuspended in 950 µL of the same buffer. Fifty µL of 1-mg/mL propidium iodide was added to a final concentration of 50 µg/mL. This was incubated at room temperature for 30 minutes. A volume of 500 µL of the cell suspension was gently vortexed and sonicated before analysis through flow cytometer (Becton Dickinson, FACS Aria SORP).

## Statistical tests

To compare different sets of data, two-tailed independent samples *t*-tests were performed. The *P*-values corresponding to four degrees of freedom were obtained to identify difference in the data due to chance.

## Whole-genome sequencing and variant calling

### Sample preparation and sequencing

Genomic DNA of ancestral haploid and evolved lines were isolated following standard zymolyase-based protocol using the kit from FAVORGEN Biotech Corporation. DNA concentrations and quality were measured using a Nano-spectrophotometer from Eppendorf and by gel electrophoresis. Samples were sent for paired-end sequencing on an Illumina HiSeq, with an average read length of 150 bp. Each of the samples had a minimum coverage of 100×.

### Mapping and variant calling

We used the cloud-based web interface system Galaxy (https://usegalaxy.eu/) to perform all sequence data analyses. Illumina paired-end reads were uploaded into the server. The quality of the reads was assessed using FastQC (version 0.72). Raw reads were trimmed with Trimmomatic (version 0.38.1) (71) and mapped to the S288c genome (version R64), and variant calling was performed using the automated tool Snippy (version 4.5.5), according to the best practice recommendations for evaluating single nucleotide variant calling (72). Variants present in the ancestral strain were filtered out manually. Finally, all remaining indels and SNPs were verified using intensive manual curation.

## ACKNOWLEDGMENTS

A.M. and S.S. thank ICTS Bangalore for the support towards attending the IV[th] Population Genetics and Evolution School, held in 2020.

We acknowledge support from DBT/Wellcome Trust (India Alliance), grant number IA/S/19/2/504632 (S.S. and P.N.), Council of Scientific and Industrial Research (CSIR), Government of India, Senior Research Fellowship (09/087(0873)/2017-EMR-I) (A.M.), and Institute Post Doctoral Fellowship Program, IIT Bombay (S.D.).

Conceptualization: A.M. and S.S.; methodology: A.M, P.N., P.V., S.D., and S.S.; investigation: A.M., P.N., P.V., S.D., and S.S.; funding acquisition: A.M., S.D., and S.S.; project administration: S.S.; supervision: A.M. and S.S.; writing: A.M. and S.S.

The authors declare that they have no competing interests related to this study.

## AUTHOR AFFILIATION

[1]Department of Chemical Engineering, Indian Institute of Technology Bombay, Mumbai, Powai, Maharashtra, India

## PRESENT ADDRESS

Anjali Mahilkar, Department of Ecology and Evolutionary Biology, University of Michigan, Ann Arbor, Michigan, USA

## AUTHOR ORCIDs

Supreet Saini http://orcid.org/0000-0001-6838-4619

## FUNDING

| Funder | Grant(s) | Author(s) |
| --- | --- | --- |
| Council of Scientific and Industrial Research, India (CSIR) | 09/087(0873)/2017-EMR-I | Anjali Mahilkar |
| The Wellcome Trust DBT India Alliance (India Alliance) | IA/S/19/2/504632 | Prachitha Nagendra |
| The Wellcome Trust DBT India Alliance (India Alliance) | IA/S/19/2/504632 | Supreet Saini |

## AUTHOR CONTRIBUTIONS

Anjali Mahilkar, Conceptualization, Data curation, Investigation, Methodology, Writing – original draft | Prachitha Nagendra, Data curation, Investigation | Pavithra Venkataraman, Data curation, Formal analysis, Methodology | Saniya Deshmukh, Data curation, Formal analysis, Investigation | Supreet Saini, Conceptualization, Funding acquisition, Methodology, Supervision, Writing – original draft, Writing – review and editing

## DATA AVAILABILITY

The raw sequencing data are available in NCBI under BioProject no. PRJNA767895.

## ADDITIONAL FILES

The following material is available online.

### Supplemental Material

**Supplemental material (Spectrum01950-23-S0001.docx).** Fig. S1 to S12 and Tables S1 to S6.

### Open Peer Review

**PEER REVIEW HISTORY (review-history.pdf).** An accounting of the reviewer comments and feedback.

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
