## [Reviewer comments · Microbiology Spectrum]

Microbiology Spectrum

Rapid evolution of pre-zygotic reproductive barriers in allopatric populations.

Anjali Mahilkar, Prachitha Nagendra, Pavithra Venkataraman, Saniya Deshmukh, and Supreet Saini

Corresponding Author(s): Supreet Saini, Indian Institute of Technology Bombay

Review Timeline:

Submission Date:	May 9, 2023
Editorial Decision:	June 1, 2023
Revision Received:	July 20, 2023
Accepted:	August 14, 2023

Editor: Kaustuv Sanyal

Reviewer(s): The reviewers have opted to remain anonymous.

Transaction Report:

DOI: <https://doi.org/10.1128/spectrum.01950-23>

June 1, 2023

Dr. Supreet Saini
Indian Institute of Technology Bombay
Chemical Engineering
Powai
Mumbai, Maharashtra 400076
India

Re: Spectrum01950-23 (Rapid evolution of pre-zygotic reproductive barriers in allopatric populations.)

Dear Dr. Supreet Saini:

Thank you for submitting your manuscript to Microbiology Spectrum. As you see below, both the reviewers appreciate the work presented but they raise some important concerns, especially one of them emphasised the need of including statistical analysis of data. When submitting the revised version of your paper, please provide (1) point-by-point responses to the issues raised by the reviewers as file type "Response to Reviewers," not in your cover letter, and (2) a PDF file that indicates the changes from the original submission (by highlighting or underlining the changes) as file type "Marked Up Manuscript - For Review Only". Please use this link to submit your revised manuscript - we strongly recommend that you submit your paper within the next 60 days or reach out to me. Detailed instructions on submitting your revised paper are below.

Link Not Available

Sincerely,

Kaustuv Sanyal

Journals Department
Reviewer comments:

Reviewer #1 (Comments for the Author):

Overall this is a strong paper about an important topic. I feel the data are interesting and merit publication. My concerns can be dealt 100% at the level of revising the writing/analysis.

Statistics - There are some very clear changes in phenotype and some less clear. I was surprised, however, not to see any statistical tests being used to support statements about improvements in growth rate or lag relative to ancestor, or between treatments. Similar for the mating efficiencies and kinetics. By eye they -seem- to be pretty clear, so I doubt this will be an issue for the conclusions, but they deserve to be added to the work.

Rationale in intro - It is stated in lines 65-69 that allopatry should establish post-zygotic and sympatric should be pre-zygotic. As this is THE central point of the paper, it would be very worthwhile to articulate what the logical rationale for each of these are (or the key studies that simply showed these to be empirical trends).

Description of evolving populations - It is confusing to read about "limiting glucose and galactose amounts" in line 83 but to later realize these are batch cultures. Unless it is at K_M or lower concentrations, then I think it was "just" glucose and galactose minimal media (usually the C is stoichiometrically less than the other components relative to growth). It was also never clear how many replicates, etc., other than inspection of the figures. A minor description of the conditions used in the beginning of the Results would be very helpful (details can of course come later in Methods).

Figure 1 - I find this a bit confusing, as movement to the right is beneficial for growth rate but it is movement down that is beneficial for lag. I'd either clarify that this is the case or perhaps flip the graph and have the y axis be hours shorter for the lag as positive values.

Mating efficiency - Can a bit more be said as to thoughts on why glu-glu pairings lose mating efficiency even faster than glu-gal?

Minor points.

Last line of the abstract is partly repetitive with above and left me unclear where mate preference pops in if only kinetics are mentioned earlier.

Line 58 - Why mention "ecological" environments? Are there "non-ecological" environments? I suggest restating to make the point clearer.

Line 102 - It is stated distinctly different and that you did the tradeoff experiments, but unlike the galactose, there was no description of what you found. This inconsistency was confusing.

Line 289 - Unclear to me why post-zygotic barriers would be rare early. I get it for the very first mutations in independent walks needing to interact well with the background, but why should they necessarily interact well with each other?

Reviewer #2 (Comments for the Author):

Adaptive response in glucose and galactose.: After 0-200 gen, increased growth rate and decreased lag phase seen for gal-evolved lines in glu, and glu-evolved lines in gal. Please comment on why such cross-adaptation might occur?

Pre-zygotic barriers to mating evolve rapidly.: Fig 2 and S2B, shouldn't number of gal-glu matings be 2x gal-gal or glu-glu?

Nature of epistasis between beneficial mutations in glucose and galactose adapted lines.: Please clarify that in yeast diploid mitotic performance is considered pre-zygotic, but in higher organisms like humans it is post-zygotic. This will prevent confusion.

Genome sequencing reveals signature of convergent evolution.: The evidence for convergent evolution is very interesting. Can the authors throw light on which mutations in Table S3 and Annexure 1 could be responsible for the higher self-mating than cross-mating frequencies?

Evolved lines exhibit altered kinetics of mating.: In Fig 5 it would have been nice to have three more controls - anc a with its mating type switched anc alpha; anc alpha with its mating type switched anc a; and anc alpha (switched) with anc a (switched). The statistical difference of the evolved haploids would then have been even more meaningful. The additional controls would not add too much extra effort.

Pre-zygotic barriers arise first in allopatric populations evolved under drift.: It would be good to show that none of the 44 drift lines exhibited signatures of convergent evolution as seen in the 12 adapted lines.

Minor point: page 10 line 217 should read "... obtained by switching their mating type ...".

Fig S4 didn't have column and row headings (maybe it didn't show up on my laptop).

Staff Comments:

Preparing Revision Guidelines

To submit your modified manuscript, log onto the eJP submission site at <https://spectrum.msubmit.net/cgi-bin/main.plex>. Go to Author Tasks and click the appropriate manuscript title to begin the revision process. The information that you entered when you

first submitted the paper will be displayed. Please update the information as necessary. Here are a few examples of required updates that authors must address:

Please return the manuscript within 60 days; if you cannot complete the modification within this time period, please contact me. If you do not wish to modify the manuscript and prefer to submit it to another journal, please notify me of your decision immediately so that the manuscript may be formally withdrawn from consideration by Microbiology Spectrum.

Comments on Mahilkar et al “Rapid evolution in allopatric populations”.

Results.

Adaptive response in glucose and galactose.: After 0-200 gen, increased growth rate and decreased lag phase seen for gal-evolved lines in glu, and glu-evolved lines in gal. Please comment on why such cross-adaptation might occur?

Pre-zygotic barriers to mating evolve rapidly.: Fig 2 and S2B, shouldn't number of gal-glu matings be 2x gal-gal or glu-glu?

Nature of epistasis between beneficial mutations in glucose and galactose adapted lines.: Please clarify that in yeast diploid mitotic performance is considered pre-zygotic, but in higher organisms like humans it is post-zygotic. This will prevent confusion.

Genome sequencing reveals signature of convergent evolution.: The evidence for convergent evolution is very interesting. Can the authors throw light on which mutations in Table S3 and Annexure 1 could be responsible for the higher self-mating than cross-mating frequencies?

Evolved lines exhibit altered kinetics of mating.: In Fig 5 it would have been nice to have three more controls – anc a with its mating type switched anc alpha; anc alpha with its mating type switched anc a; and anc alpha (switched) with anc a (switched). The statistical difference of the evolved haploids would then have been even more meaningful. The additional controls would not add too much extra effort.

Pre-zygotic barriers arise first in allopatric populations evolved under drift.: It would be good to show that none of the 44 drift lines exhibited signatures of convergent evolution as seen in the 12 adapted lines.

Minor point: page 10 line 217 should read "... obtained by switching their mating type ...".

Fig S4 didn't have column and row headings (maybe it didn't show up on my laptop).

Reviewer #1 (Comments for the Author):

Overall this is a strong paper about an important topic. I feel the data are interesting and merit publication. My concerns can be dealt 100% at the level of revising the writing/analysis.

Statistics - There are some very clear changes in phenotype and some less clear. I was surprised, however, not to see any statistical tests being used to support statements about improvements in growth rate or lag relative to ancestor, or between treatments. Similar for the mating efficiencies and kinetics. By eye they -seem- to be pretty clear, so I doubt this will be an issue for the conclusions, but they deserve to be added to the work.

Response: We have now added the statistical comparisons between the changes in growth rate and lag phase duration in the edited version of the manuscript. (Figure 1 and Table S1 in the revised manuscript).

Rationale in intro - It is stated in lines 65-69 that allopatry should establish post-zygotic and sympatric should be pre-zygotic. As this is THE central point of the paper, it would be very worthwhile to articulate what the logical rationale for each of these are (or the key studies that simply showed these to be empirical trends).

Response: The statements made here refer to exhaustive empirical studies by Coyne & Orr (1989, 1997). We now cite this reference against the statement. We have also provided another reference from a laboratory study in yeast (Dettman et al, 2008, *Nature*) where the authors show that post-zygotic barrier evolves in yeast populations in allopatry. From the context of laboratory studies studying incipient speciation with isogenic populations, the nature of barriers that arise first remains an open question. (Lines 66-68 in the revised manuscript).

Description of evolving populations - It is confusing to read about "limiting glucose and galactose amounts" in line 83 but to later realize these are batch cultures. Unless it is at K_M or lower concentrations, then I think it was "just" glucose and galactose minimal media (usually the C is stoichiometrically less than the other components relative to growth). It was also never clear how many replicates, etc., other than inspection of the figures. A minor description of the conditions used in the beginning of the Results would be very helpful (details can of course come later in Methods).

Response: We now provide the following details in the introduction section. We have also edited the text to refer to the environments as "glucose-limited" and "galactose-limited". (Lines 83-88 of the revised manuscript)

"Haploid isogenic yeast populations of both mating types were evolved in glucose- and galactose-limited growth media for 600 generations, respectively. Three MAT_a and MAT_α lines were evolved in glucose; and three MAT_a and MAT_α lines were evolved in galactose – thus, making a total of 12 allopatric populations in the experiment (**Fig. S1**). To study evolution in the absence of selection, 44 independent lines of haploids (22 lines of MAT_a and 22 lines of MAT_α) were also evolved in a mutation accumulation experiment for 70 transfers (~1500 generations)."

Figure 1 - I find this a bit confusing, as movement to the right is beneficial for growth rate but it is movement down that is beneficial for lag. I'd either clarify that this is the case or perhaps flip the graph and have the y axis be hours shorter for the lag as positive values.

Response: We have now flipped the y-axis as per the reviewer's suggestion. In the edited representation, the y-axis stands for "Reduction in lag phase duration (hours)". Positive values in this case represent an increase in fitness, while negative values (an increase in lag phase duration) represent a decrease in fitness. (Figure 1 of the revised manuscript)

Mating efficiency - Can a bit more be said as to thoughts on why glu-glu pairings lose mating efficiency even faster than glu-gal?

Response: *S. cerevisiae*'s adaptive trajectory in glucose is qualitatively different from its adaptation in other carbon environments. The species, due to its evolutionary origins and trajectory, has accumulated 16 proteins dedicated to transport of glucose from the environment. As a comparison, *S. cerevisiae* has only one transporter for bringing in extracellular galactose.

Kvitek and Sherlock, in 2011 *PLoS Genetics*, reported that when two mutations, both individually beneficial in a glucose environment when brought together in the same genome lead to a decrease in fitness of the strain. The molecular mechanism of this phenomenon is not known.

Our data also suggests the same (Figure 4A). This is quite unlike what is observed in galactose (Figure 4B).

Thus, even from the context of adaptation, it is apparent that adaptation in glucose is unlike that in another environment. The implications of this phenomenon on speciation or evolution of reproductive barriers is not known.

As a result, we feel that we are unable to comment on the phenomenon of glu-glu pairings losing mating efficiency faster compared to glu-gal. In fact, investigation of this phenomenon is a major emphasis of one of our current investigations.

Minor points.

Last line of the abstract is partly repetitive with above and left me unclear where mate preference pops in if only kinetics are mentioned earlier.

Response: We have now removed the repetitive line in the abstract. We thank the reviewer for pointing this out.

Line 58 - Why mention "ecological" environments? Are there "non-ecological" environments? I suggest restating to make the point clearer.

Response: We agree with the reviewer's critique and have dropped the reference to the environment in the statement. (Line 56 in the revised manuscript)

Line 102 - It is stated distinctly different and that you did the tradeoff experiments, but unlike the galactose, there was no description of what you found. This inconsistency was confusing.

Response: We thank the reviewer for this comment. We have now expanded the section and added explicit descriptions of all our results. (Lines 108-117 in the revised manuscript)

Line 289 - Unclear to me why post-zygotic barriers would be rare early. I get it for the very first mutations in independent walks needing to interact well with the background, but why should they necessarily interact well with each other?

Response: We agree with the critique of the reviewer, and we have now edited the last paragraph of the discussion section. (Lines 312-315 of the revised manuscript).

Additionally, independent mutations in adaptive walks have been found to be beneficial together too (Khan et al, 2011 *Science*). The exception to this is when the beneficial mutations are in the same gene (Weinreich et al, 2006 *Science*). Another scenario when beneficial mutations do not confer benefit together has been observed when allopatric populations adapt to glucose (Kvitek and Sherlock, 2011 *PLoS Genetics*).

Reviewer #2 (Comments for the Author):

Adaptive response in glucose and galactose.: After 0-200 gen, increased growth rate and decreased lag phase seen for gal-evolved lines in glu, and glu-evolved lines in gal. Please comment on why such cross-adaptation might occur?

Response: Cross-adaptation during growth on carbon substrates has been reported several times (Choudhury and Saini, 2019 *JEB*; Chen and Zhang, 2020 *NEE*). This is likely due to the fact that increased efficiency of glycolysis enhance growth rates for carbon sources which feed intermediates into the pathway. We make a note of this in the revised manuscript. (Lines 114-117 in the revised manuscript)

Pre-zygotic barriers to mating evolve rapidly.: Fig 2 and S2B, shouldn't number of gal-glu matings be 2x gal-gal or glu-glu?

Response: Our mating experiments between the glucose and galactose adapted lines were performed between the MAT α galactose-evolved lines and MAT α glucose-evolved lines. Hence, the number of mating experiment curves is 9 (and not 18). We have now explicitly mentioned this in the revised manuscript. (Line 18 in the Supplement)

Nature of epistasis between beneficial mutations in glucose and galactose adapted lines.: Please clarify that in yeast diploid mitotic performance is considered pre-zygotic, but in higher organisms like humans it is post-zygotic. This will prevent confusion.

Response: Diploid growth rate in yeast is a measure of a post-zygotic isolation. The only pre-zygotic measure of reproductive isolation in yeast is the efficiency with which the participating haploids mate with each other.

Genome sequencing reveals signature of convergent evolution.: The evidence for convergent evolution is very interesting. Can the authors throw light on which mutations in Table S3 and Annexure 1 could be responsible for the higher self-mating than cross-mating frequencies?

Response: We are unable to speculate at this point regarding the identity of mutations which are responsible for higher self-mating than cross-mating. In fact, the identification of these mutations and the molecular mechanisms underlying these changes is currently a large part of the effort from our group. We anticipate that the reviewer's comment will form the basis of several next manuscripts from our group.

Evolved lines exhibit altered kinetics of mating.: In Fig 5 it would have been nice to have three more controls - anc a with its mating type switched anc alpha; anc alpha with its mating type switched anc a; and anc alpha (switched) with anc a (switched). The statistical difference of the evolved haploids would then have been even more meaningful. The additional controls would not add too much extra effort.

Response: We have now performed these control experiments, and reported them in the Figure S8 (Page 9 of the Supplement) in the revised manuscript.

Pre-zygotic barriers arise first in allopatric populations evolved under drift.: It would be good to show that none of the 44 drift lines exhibited signatures of convergent evolution as seen in the 12 adapted lines.

Response: We are currently unable to sequence the 44 lines evolved in the mutation accumulation experiment. However, our null expectation is that since these 44 lines are evolved under drift, and selection is negligible, the molecular targets of mutational events will be unrelated to adaptation in the environmental context in which these lines have been propagated. This phenomenon has been recently reported in the mutations identified in an exhaustive mutation accumulation experiment in bacteria (Sane et al, 2023 *PNAS*).

We thank the reviewer for this comment and have expanded the discussion on this aspect in the edited manuscript. (Lines 265-269 in the revised manuscript)

Minor point: page 10 line 217 should read "... obtained by switching their mating type ...".

Response: We have now edited the text as per the reviewer's suggestion.

Fig S4 didn't have column and row headings (maybe it didn't show up on my laptop).

Response: We request the reviewer to kindly check the Figure S4 again. In our version of the manuscript, the rows and columns are labelled as α and a , respectively.

August 14, 2023

Dr. Supreet Saini
Indian Institute of Technology Bombay
Chemical Engineering
Powai
Mumbai, Maharashtra 400076
India

Re: Spectrum01950-23R1 (Rapid evolution of pre-zygotic reproductive barriers in allopatric populations.)

Dear Dr. Supreet Saini:

Your manuscript has been accepted, and I am forwarding it to the ASM Journals Department for publication. You will be notified when your proofs are ready to be viewed.

Sincerely,

Kaustuv Sanyal
Editor, Microbiology Spectrum
